# The Positive Regulatory Effect of DBT on Lipid Metabolism in Postpartum Dairy Cows

**DOI:** 10.3390/metabo15010058

**Published:** 2025-01-16

**Authors:** Zheng Zhou, Kang Yong, Zhengzhong Luo, Zhenlong Du, Tao Zhou, Xiaoping Li, Xueping Yao, Liuhong Shen, Shumin Yu, Yixin Huang, Suizhong Cao

**Affiliations:** 1Department of Clinical Veterinary Medicine, College of Veterinary Medicine, Sichuan Agricultural University, Chengdu 611130, China; fultz20zzheng@163.com (Z.Z.); dzl960730@163.com (Z.D.); zhoutao0428@163.com (T.Z.); 13577@sicau.edu.cn (X.Y.); shenlh@sicau.edu.cn (L.S.); yayushumin@sicau.edu.cn (S.Y.); 2Key Laboratory of Animal Disease and Human Health of Sichuan Province, College of Veterinary Medicine, Sichuan Agricultural University, Chengdu 611130, China; 3Department of Animal Husbandry and Veterinary Medicine, College of Animal Science and Technology, Chongqing Three Gorges Vocational College, Chongqing 404105, China; yongkangkang@126.com; 4Department of Clinical Veterinary Medicine, College of Veterinary Medicine, China Agricultural University, Beijing 100193, China; xiaopingli0512@163.com

**Keywords:** DBT, lipid metabolism, hindgut microbiota, metabolomics, bile acids

## Abstract

Background/Objectives: The transition from a non-lactating to a lactating state is a critical period for lipid metabolism in dairy cows. Danggui Buxue Tang (DBT), stimulating energy metabolism, ameliorates diseases related to lipid metabolism disorders and is expected to be an effective supplement for alleviating excessive lipid mobilisation in periparturient dairy cows. This study aimed to investigate the effects of supplemental DBT on serum biochemical indices, faecal microbial communities, and plasma metabolites in dairy cows. Methods: Thirty cows were randomly divided into three groups: H-DBT group, L-DBT group, and control group. DBT administration was started on the day of calving and continued once daily for seven days. Faecal and blood samples were collected on calving day, 7 days after calving, and 14 days after calving. The levels of serum biochemical indices were measured at three time points in the three groups using commercial kits. Cows in the H-DBT group and control group were selected for metabolome and 16S rRNA amplicon sequencing. Results: Our research shows that, in dairy cows 7 days postpartum, DBT significantly reduced serum 3-hydroxybutyric acid (BHB) concentrations and the number of cows with BHB concentrations ≥ 1 mmol/L. Additionally, DBT increased serum total cholesterol contents at both 7 and 14 days postpartum. Analysis of the microbiota community showed that DBT modulated the composition and structure of the hindgut microbiota. Metabolomic analysis revealed decreased plasma acetylcarnitine, 2-hydroxybutyric acid, and BHB levels 7 days postpartum, whereas the TCA cycle was enhanced. At 14 days postpartum, DBT altered the plasma bile acid profile, especially glycine-conjugated bile acids, including GCDCA, GUDCA, and GDCA. Correlation analyses showed that the relative abundances of *Bacillus*, *Solibacillus, Dorea*, and *Romboutsia* were strongly correlated with the differential metabolites, which is crucial for the beneficial effects of DBT. Conclusions: DBT improves energy status and lipid metabolism in postpartum dairy cows by modulating hindgut microbiota and serum lipid metabolites.

## 1. Introduction

The periparturient period, defined as three weeks before and after calving [1], is critical in the dairy cow production cycle and can be challenging for both cows and producers. During this period, cows experience pregnancy, calving, lactation, uterine recuperation, and endocrine and metabolic state changes while facing nutritional stress caused by a decrease in dry matter intake and an increase in energy demand, bringing the cows into a physiologically unavoidable state of negative energy balance (NEB; [2,3]). The development of NEB leads to excessive lipid mobilisation, particularly fatty acids from the adipose tissue, which eventually become ketone bodies and cause metabolic diseases, such as hyperketonaemia and fatty liver in dairy cows [4,5]. Bile acids (BA), as cholesterol-derived molecules, are signalling molecules involved in the regulation of glucose, lipid, and energy metabolism as well as in facilitating lipid absorption [6]. Accumulating evidence demonstrates that the BA have an essential role in lipid metabolism in periparturient dairy cows [7,8]. Previous studies showed that NEB and lipid mobilisation are more severe in multiparous than in primiparous cows [9,10]. Therefore, it is crucial to adopt appropriate management strategies to adapt cows to this new physiological state to meet their energy needs. Nutritional supplementation is one of the most researched management strategies for managing transition-related stress [11]. Only when a cow smoothly passes through the periparturient period can a solid foundation be laid for high production throughout the lactation cycle.

Danggui Buxue Tang (DBT) is a classic prescription that was first recorded by Dongyuan Li in ‘*Neiwaishangbianhuolun Shushangweiqilun*’ 1247 CE [12]. It is composed of Huangqi (HQ, Astragali Radix) and Danggui (DG, Angelica Sinensis Radix) at a 5:1 ratio [13]; HQ and DG are both medicinal and edible herbs, and studies have shown that DBT has the best chemical and biological properties when HQ and DG are mixed at this ratio [14,15]. According to ancient traditional Chinese medicine (TCM) theory, DBT mainly supports ‘Qi’ and enriches the blood [16]. In TCM, Qi is commonly accepted as the vital energy used in constructing the human body, maintaining living activities [17], and stimulating vital energy/energy metabolism [12]. Furthermore, modern pharmacological studies have shown that DBT and its components have pharmacological benefits, such as anti-fatigue, anti-inflammation, and regulation of immune abnormalities and menopausal symptoms [12,18,19]. It has been shown that DBT improves lipid metabolism in diabetic mice [20,21], suggesting its potentially pivotal role in regulating lipid metabolism.

The intestinal microbiota is essential for the metabolic health of the host. However, when dysregulated, it may contribute to the pathogenesis of various metabolic diseases, including type 2 diabetes mellitus, obesity, and non-alcoholic liver disease [22]. These diseases are associated with lipid metabolism disorders, and the intestinal flora is crucial for maintaining homeostasis of host lipid metabolism by regulating the absorption, metabolism, and storage of mammalian lipids [23]. Studies suggest that gut microbiota dysbiosis contributes to disease development in dairy cows, including mastitis and diarrhoea [24,25]. The gut microbial composition is altered in cows with metabolic disorders, such as left-displaced abomasum and hyperketonaemia [26,27]. In addition, the gut flora is a core component of TCM studies, and the interaction between intestinal flora and TCM is critical for host health [28]. Researchers have shown that DBT can exert a therapeutic effect on diseases by affecting the intestinal flora. Wang et al. [29] showed that DBT restores gut microbiota imbalance and regulates metabolic pathways in the gut microbiota of type 2 diabetic mice. Du et al. [30] reported that DBT restores the abundance of *Bacteroides* and *Rikenellaceae_RC9_Gut_group* and partially restores the intestinal flora and metabolic disorders caused by antibiotics.

The efficacy of DBT is reduced when antibiotics disrupt the gut flora [30]. In addition, DBT significantly improves the immune response to intestinal mucosal imbalance, rebuilds the intestinal mucosal barrier, improves inflammatory bowel disease, and restores the intestinal microecological balance [18]. Metabolomics is a holistic investigation based on global metabolic profiles in complex biological matrices. It is the process of detecting changes in small-molecule metabolites and tapping them into differential metabolic pathways in the body [31]. In agreement with the holistic thinking of TCM, metabolomics has the potential to evaluate the efficacy and biochemical action mechanisms of TCM [32].

Based on the pharmacological effects of DBT and TCM theory, we hypothesised that DBT may improve NEB and alleviates lipid mobilisation in postpartum dairy cows. However, relevant experiments or literature reports are lacking. Thus, this study aimed to analyse the effects of DBT supplementation on postpartum dairy cows and explore the role of the DBT-derived metabolome and microbiome in the benefits of DBT, both horizontally (in different groups) and longitudinally (at different times). This study lays a theoretical foundation for the scientific addition of DBT during the periparturient period, providing new concepts for correcting lipid metabolism disorders in periparturient cows.

## 2. Materials and Methods

### 2.1. Preparation of DBT

HQ and DG were purchased from Minxian Ronghe Pharmaceutical Co., Ltd. (Dingxi, Gansu, China). Each herb was air-dried and ground in a mixer to a fine powder (particle size < 15 µm). Fine powders of HQ and DG were dissolved in 3000 mL of hot water at a ratio of 5:1 and allowed to cool. According to ‘Pharmacopoeia of the People’s Republic of China 2020 Edition’ and literature reports [30], formononetin and ferulic acid were selected as quality markers of DBT and analysed using high-performance liquid chromatography–mass spectrometry. The relative peak area percentages of formononetin and ferulic acid were 0.235 and 0.045%, respectively.

### 2.2. Animal Grouping, Management, and Sample Collection

Animal samples were collected per the Guidelines for the Care and Use of Laboratory Animals of China. All procedures were approved by the Animal Care and Use Committee of Sichuan Agricultural University (No. 20220163, 2 March 2022). The experiments were conducted at a commercial dairy farm in the Ningxia Hui Autonomous Region, China, from December to May 2023. During the trial period, the cows were uniformly managed and fed a total mixed ration (TMR) thrice daily with free access to food and water. Dry matter intakes (DMI) were calculated based on the amounts of feed offered and refused together with the DM content of the TMR. There was no difference in average DMI among groups. Thirty multiparous Chinese Holstein dairy cows (2–3 parity) with similar body condition scores, actual days of gestation, calf weights, or calving ease scores and without clinical diseases were randomly selected. Thirty cows were randomly divided into three groups (10 cows in each group): high-dose DBT group (H-DBT group), low-dose DBT group (L-DBT group), and control group. Cows in the H-DBT group received a daily dose of 240 g DBT (200 g HQ, 40 g DG), cows in the L-DBT group received a daily dose of 120 g DBT (100 g HQ, 20 g DG), and cows in the control group received 3000 mL of water per day like the others. Treatments were administered via esophageal tubing inserted directly into the rumen through a stainless-steel speculum. The placement of the distal end of the tubing in the rumen liquor was confirmed by blowing on the proximal end of the tubing and listening for gas bubbles using a stethoscope, ensuring accurate delivery of the treatment directly into the rumen. DBT administration was started on the day of calving and continued for seven days (10:00 a.m. every day).

Faecal and blood samples were collected at three time points: the day of calving (0 days), 7 days after calving (+7 days), and 14 days after calving (+14 days). Each sample was collected before the morning feeding. The control and H-DBT groups on 0 days were named C0 and H-DBT0, respectively. The control and H-DBT groups on +7 days were named C7 and H-DBT7, respectively. The control and H-DBT groups on +14 days were named C14 and H-DBT14, respectively. Blood BHB concentrations were detected on days 0, 7, and 14 using a NovaVat Blood Ketone Meter (WD1621; Nova Bio Vet, Waltham, MA, USA). Serum and plasma samples were collected (using heparin sodium as an anticoagulant), centrifuged at 1500× *g* for 10 min at 25 °C, and stored at −80 °C. Faecal samples were collected via the rectum using disposable sterile long arm gloves, transferred into freezing tubes, and stored at −80 °C. Serum samples from the three time points and groups were tested for biochemical indices. The optimal dose was determined based on clinical phenotypes. 16S rRNA amplicon sequencing and targeted metabolomics were performed on the optimal dose and control groups at different times.

### 2.3. Measurement of Serum Biochemical Indices

Serum glucose (Glu), non-esterified fatty acids (NEFA), triglyceride (TG), total cholesterol (T-CHO), total protein (TP), albumin (ALB), and GSK3B concentrations were determined using commercially available kits (Nanjing Jiancheng Bioengineering Institute, Nanjing, China; nos. A154-1-1, A042-2-1, A110-1-1, A111-1-1, A045-2-2, A028-1-1, and H572-1-2, respectively). GLU was measured using the glucose oxidase method; NEFA was determined by enzymatic assay; TG was measured using the GPO-PAP enzymatic method; T-CHO was quantified via the COD-PAP enzymatic method; TP was assessed using the BCA microplate method; ALB was measured using microwell plate method; and GSK3B was measured using a competitive method (antigen–antibody binding).

### 2.4. Faecal 16S rRNA Amplicon Sequencing and Data Processing

Total DNA was extracted from dairy cow stool samples using the FastPure Faeces DNA Isolation Kit (Vazyme, Nanjing, China). The hypervariable V3–V4 region of the bacterial 16S rRNA gene was sequenced using primers (341F: 5′-CCTAYGGGRBGCASCAG-3′; 806R: 5′-GGACTACNNGGGTATCTAAT-3′) and a T100 Thermal Cycler PCR thermocycler (BIO-RAD, Hercules, CA, USA). PCR amplification cycling conditions were as follows: initial denaturation at 95 °C for 3 min, followed by 27 cycles of denaturation at 95 °C for 30 s, annealing at 55 °C for 30 s, and extension at 72 °C for 45 s, with a final extension at 72 °C for 10 min, followed by cooling to 4 °C. The PCR product was extracted from a 2% agarose gel, purified using the PCR Clean-Up Kit (YuHua, Shanghai, China) according to the manufacturer’s instructions, and quantified using a Qubit 4.0 (Thermo Fisher Scientific, Waltham, MA, USA). Sequences obtained after demultiplexing were quality-filtered using fastp (https://github.com/OpenGene/fastp, 15 March 2024, version 0.19.6) and merged with FLASH (http://www.cbcb.umd.edu/software/flash, 15 March 2024, version 1.2.11). After length filtering and denoising, amplicon sequence variants (ASVs) were obtained using DADA2 in QIIME2 (https://qiime2.org, 15 March 2024). Species taxonomic analysis of ASVs was performed using the SILVA 16S rRNA database (https://www.arb−silvac.de, 15 March 2024, v138).

### 2.5. Targeted Analysis of Plasma Metabolites and Data Processing

Sample preparation was first performed, followed by an ultra-performance liquid chromatography–tandem mass spectrometry (UPLC–MS/MS) analysis to quantify all targeted metabolites (ACQUITY UPLC-Xevo TQ-S, Waters Corp., Milford, MA, USA). Specific sample preparation methods, details on the mobile phase and gradient elution procedure for UPLC, and mass spectrometry conditions are described in the Appendix A. A total of 140 metabolites were detected, including amino acids (35), fatty acids (25), bile acids (23), organic acids (18), and others (39). The differential metabolites were visualised, and the Kyoto Encyclopedia of Genes and Genomes (KEGG) pathway analysis was performed using MetaboAnalyst 5.0.

### 2.6. Statistical Analysis

Statistical analyses were performed using SPSS software (version 27.0; IBM SPSS Inc., Chicago, IL, United States). Between-group *p*-values were calculated using a one-way analysis of variance (ANOVA) based on conformity to normal distribution. The significance threshold was set to *p* < 0.05; trends were declared at 0.05 ≤ *p* < 0.10. Data are presented as mean ± standard error unless otherwise indicated. Chi-square was performed based on the number of cows with serum concentrations BHB ≥ 1 mmol/L.

Regarding faecal 16S rRNA amplicon sequencing, the Shannon and Chao 1 indices represent the α diversity of the two comparison groups. The variability in α diversity was reflected by the *p*-value of the Wilcoxon rank-sum test. The β diversity of the two comparison groups was obtained based on Bray–Curtis distances. The variability in the β diversity was illustrated by *p*-values based on the Adonis analysis with 999 permutations. Different phyla and genera were screened using the Wilcoxon rank-sum test. For targeted metabolomic data, if the data of a metabolite met normal distribution and homogeneity, the metabolite was compared between groups using the *t*-test; otherwise, the metabolite was compared between groups using the Mann–Whitney U-test to obtain the differential metabolite. Finally, Spearman’s correlation coefficient was computed using the R software 4.3.3 to analyse the relationship between metabolites and microbes.

## 3. Results

### 3.1. Effect of DBT on Serum Biochemical Indices in Postpartum Dairy Cows

The effects of DBT on the blood biochemical indicators of postpartum dairy cows are illustrated in Figure 1. The results indicate that the H-DBT7 group exhibited significantly reduced BHB values compared with those of the control group (*p* < 0.05); moreover, a trend towards lower BHB values was observed for the L-DBT7 group (0.05 < *p* < 0.1). Serum TG levels were significantly increased in the L-DBT7 group (*p* < 0.05) and increased in the H-DBT7 group (0.05 < *p* < 0.1). Serum GLU levels in the L-DBT14 group were significantly elevated compared to the control group (*p* < 0.05). Serum T-CHO levels were higher in the H-DBT14 (*p* < 0.01) and L-DBT14 (*p* < 0.05) groups than those in the control group. However, no significant change was observed in the NEFA, TP, ALB, and ALB to TP ratios in cows in all three groups at the three time points. We performed a chi-square test based on the number of cows with serum BHB concentrations ≥ 1 mmol/L. According to Table 1, χ^2^ = 7.500 and *p* = 0.006 < 0.0167 for the H-DBT group versus the control group, suggesting a relationship between high doses of DBT and the number of cows with serum BHB contents ≥ 1 mmol/L. In contrast, 14 days after calving, no statistically significant differences between the groups were observed (Table 2). Therefore, the control and H-DBT groups were selected for subsequent omics analyses.

### 3.2. DBT-Modulated Intestinal Microbiota

Intestinal flora represents the core of TCM studies. By analysing faecal 16S rRNA in late periparturient dairy cows, the effect of DBT on the gut microbial communities was determined. The richness and evenness indices of faecal microbiota, including Shannon and Chao1 diversity indices, did not significantly differ between the control and H-DBT groups at any time point (Figure 2A). However, principal coordinate analysis (PCoA) showed a clear separation between the C14 and H-DBT14 groups (*p* = 0.005; Figure 2B), indicating that DBT did not significantly change the bacterial diversity but altered the overall microbiota structure.

We further analysed the composition and structure of faecal microbial populations at the phylum and genus levels. According to the phylum assignment results, Firmicutes and Bacteroidetes were the dominant bacteria, accounting for >90% of the taxonomic groups identified at all time points (Figure 2C). In addition, compared with that of the C14 group, the relative abundance of Bacteroidetes significantly increased in the H-DBT group (*p* < 0.05), whereas the ratio of Firmicutes to Bacteroidetes (F/B) significantly decreased in the H-DBT group (*p* < 0.05; Figure 2D). According to the genus assignment results, the dominant genera were *UCG-005* and *Rikenellaceae-RC9-gut-group* (Figure 2E). Compared with that of the C7 group, the relative abundance of *UCG-009* notably decreased in the H-DBT group. In contrast, those of *Psychrobacillus*, *Bacillus*, *Solibacillus*, *Arthrobacter*, and *Aerococcus* significantly increased in the H-DBT group (Figure 2F). Eighteen differential genera were identified between the C14 and DBT14 groups, among which the top 10 differential species are shown in Figure 2G.

To explore the correlation between phenotypic indicators and the microbiome, Spearman’s correlation analyses were performed on all phenotypic indicators and microbiome data at the three time points (Figure 3A). The results showed that the relative abundance of Verrucomicrobiota was significantly positively correlated with glucose (R = 0.5, *p* < 0.001; Figure 3F) and significantly negatively correlated with BHB (R = −0.56, *p* < 0.001; Figure 3B) and T-CHO (R = −0.56, *p* < 0.001; Figure 3D). The relative abundance of Actinobacteriota showed a significant negative correlation with Glu (R = −0.45, *p* < 0.001; Figure 3G) and a positive correlation with plasma BHB (R = 0.57, *p* < 0.001; Figure 3C) and T-CHO (R = 0.36, *p* < 0.01; Figure 3E).

### 3.3. DBT Changed Plasma Metabolic Profiles

Metabolomics allows for qualitative and quantitative analyses of host small-molecule metabolites. Plasma-targeted metabolomic analysis was conducted using UPLC–MS/MS to screen differential metabolites after DBT infusion. Based on the comparison of serum metabolomics with controls 7 days postpartum, 16 differential metabolites (*p* < 0.1) were identified, among which 12 were considered to be significantly expressed (*p* < 0.05). Fourteen days postpartum, 15 differential metabolites (*p* < 0.1) were identified, among which eight were considered significantly expressed (*p* < 0.05).

Compared with those of the C7 group, the contents of 4-aminobutanoate (GABA), fumaric acid, malic acid, N-acetylserotonin, ornithine, and aspartic acid (Asp) were significantly increased in the DBT7 group (*p* < 0.05). In contrast, the contents of 2-phenylpropionate, hydrocinnamic acid, BHB, 2-hydroxybutyric acid (2-HB), acetylcarnitine, and ethylmethylacetic acid were significantly reduced in the H-DBT group (*p* < 0.05; Figure 4A). The differential metabolite heatmap shows the distribution and classification of differential metabolites between groups (Figure 4C). To further explore the effects of DBT, KEGG pathway analysis was performed. The pathway analysis of the 12 significantly altered metabolites in the KEGG ‘Bos taurus’ database is shown in Figure 4E. Five of the metabolic pathways had *p*-values below 0.05, including arginine and proline metabolism (bta00330); alanine, aspartate, and glutamate metabolism (bta00250); citrate cycle (TCA cycle; bta00020); butanoate metabolism (bta00650); and synthesis and degradation of ketone bodies (bta00072; Figure 4F).

Compared with those of the C14 group, asparagine, 3β-ursodeoxycholic acid (bUDCA), glycoursodeoxycholic acid (GUDCA), 2-hydroxy-3-methylbutyric acid, glycodeoxycholic acid (GDCA), 3-hydroxyhippuric acid, 4-aminohippuric acid, and glycochenodeoxycholate (GCDCA) contents were significantly reduced in the H-DBT group (*p* < 0.05; Figure 4B). In this classification, four differential metabolites belonged to the bile acid (BA) class (Figure 4D). Significantly altered differential metabolites were subjected to KEGG enrichment analysis, and the pathways were enriched with *p*-values greater than 0.05.

Subsequently, we queried differential metabolites in the KEGG pathway database and searched relevant literature for data relating to the overall metabolism pathway (Figure 5). These results suggest that DBT altered the TCA cycle and lipid metabolism in late periparturient dairy cows.

### 3.4. Correlation Among Phenotypes, Genera, and Plasma Metabolites

Spearman’s correlation analysis was used to further examine the correlations among phenotype, differential serum metabolites, and differential genera to gain a holistic view of DBT effects. First, differential metabolites were subjected to Spearman’s correlation analysis with the relative abundance of differential genera 7 days postpartum. The results showed that acetylcarnitine, 2-HB, and Asp were significantly correlated with several differential genera (Figure 6A). Particularly, acetylcarnitine was significantly negatively correlated with *Psychrobacillus* (R = −0.56, *p* < 0.001), *Bacillus* (R = −0.39, *p* = 0.013), and *Solibacillus* (R = −0.53, *p* < 0.001); 2-HB showed a significant negative correlation with *Bacillus* (R = −0.38, *p* = 0.015) and *Solibacillus* (R = −0.33, *p* = 0.04); and Asp exhibited a significant positive correlation with *Bacillus* (R = 0.35, *p* = 0.026) and *Solibacillus* (R = 0.36, *p* = 0.024; Figure 6C). We then visualised the correlations between the differential metabolites and genera at 7 days postpartum using a correlation network diagram (Appendix A). Acetylcarnitine showed strong negative correlations with most microbial taxa, whereas BHB displayed significant negative correlations with *Bacillus* and *Solibacillus*.

We continued our Spearman correlation analyses of differential metabolites and flora 14 days postpartum. The results showed that *Dorea* and *Romboutsia* were significantly correlated with several bile acids (Figure 6B). *Dorea* showed a significant negative correlation with GCDCA (R = −0.35, *p* = 0.026), GDCA (R = −0.4, *p* = 0.011), and GUDCA (R = −0.35, *p* = 0.025). *Romboutsia* showed a significant positive correlation with GCDCA (R = 0.42, *p* < 0.01), GDCA (R = 0.4, *p* = 0.011), and GUDCA (R = 0.45, *p* < 0.01; Figure 6D).

To further analyse the correlation between blood metabolites and the microbial community structure, we performed network association analyses between microbial variables (genus level) and blood metabolites (Figure 6E). We observed that BHB levels were strongly correlated with the community structures of the DBT7, DBT14, and C14 groups, and 2-HB and 3-hydroxyhippuric acid levels were closely related to the community structure of the C7 group.

### 3.5. DBT Intervention Alters the Serum BA Profiles of Postpartum Dairy Cows

Twenty-three types of BA were identified in postpartum dairy cows (six primary and seventeen secondary). We observed that the differential metabolites at 14 days after calving predominantly belonged to the BA class, and the differential genera also correlated strongly with BA. However, the differential metabolites at 7 days postpartum were not associated with BA. We, therefore, analysed BA profiles at +14 days.

In addition to the four significantly different BA mentioned above, a trend towards lower GCA values was observed for the DBT14 group (0.05 < *p* < 0.1) (Figure 7A). The total bile acid (TBA), primary bile acid (PBA), and secondary bile acid (SBA) levels in the plasma demonstrate a decreasing tendency in the DBT group (Figure 7B). Similarly, 12-OH and non-12-OH BA also show a decreasing tendency (Figure 7C). The 23 BA tested were categorised according to their source and structure into unconjugated primary BA (UnconPBA), conjugated primary BA (conPBA), unconjugated secondary BA (UnconSBA), and conjugated secondary BA (conSBA). The major BA in the plasma was conPBA, followed by UnconPBA (Appendix A). However, there was no significant change in the percentage of these four BAs. Since we observed that the differential BAs were mainly focused on glycine-conjugated BA (GBA), we analysed the five GBAs and found that GBA was significantly lower in the DBT group (*p* < 0.05). Finally, we performed a correlation analysis between the bile acid profiles and phenotypic indicators at postpartum days 7 and 14 (Figure 7E). The results showed that BHB was significantly and positively correlated with GCA, GDCA, TCA, and GCDCA.

## 4. Discussion

This study aimed to explore the role of DBT in post-parturient cows and provide a theoretical basis for the scientific addition and clinical application of DBT during the periparturient period of dairy cows. Among phenotypic indices, the plasma BHB concentration indicates excessive postpartum NEB [33]. Threshold values for interpreting BHB were based on previous research; BHB ≥ 1 mmol/L adversely affects reproductive performance and milk production and is associated with an increased risk of subsequent diseases [34,35], and BHB ≥ 1 mmol/L has been used as a marker of NEB in postpartum period dairy cows [36,37]. It has been shown that an early hyperketonaemia diagnosis (within one week postpartum) is associated with a higher risk of negative outcomes than a diagnosis in the second week of lactation or later [38,39]. Mann and McArt [40] noted that an increase in blood BHB levels during the first week of lactation indicates an underlying physiological issue. Therefore, hyperketonaemia during the first week of lactation indicates that cows are not well adapted to the initiation of lactation. Low plasma T-CHO concentrations are characteristics of NEB in early lactating dairy cows. Gross et al. [41] suggested that higher cholesterol concentrations in early lactation improve hepatic TG export and alleviate adaptation to metabolic stress during increased lipomobilisation. In this study, we investigated the effects of DBT on cows during the late peripartum period. After 1 week of DBT treatment, we observed that the serum BHB levels in the H-DBT group were significantly reduced compared with those of the control group; the number of individuals with BHB ≥ 1 decreased (7 days), and serum T-CHO levels increased (7 and 14 days). Our data suggest that DBT can help postpartum cows better adapt to the lactation phase, alleviate the NEB status, and regulate lipid metabolism in cows during early lactation and that the H-DBT group is more effective than the L-DBT group.

Gut microbiota is crucial for the health and metabolic diseases of dairy cows [42], and fermentation processes in the hindgut provide an essential energy source for cows [43]. The results of our previous study show that dynamic changes in faecal microbial taxonomic taxa in periparturient dairy cows are associated with metabolic adaptations and that abnormalities in faecal microbial community composition contribute to the development of hyperketonaemia [44]. Based on our experimental data, the relative abundance of Verrucomicrobiota strongly positively correlates with glucose levels and strongly negatively correlates with BHB and T-CHO. Verrucomicrobiota is thought to regulate Glu homeostasis, which is consistent with core metabolic pathways in the functional annotations of Verrucomicrobiota [45,46]. In contrast, it has been shown that Actinobacteriota is involved in the biodegradation of resistant starch, and its abundance is positively correlated with a high-fat diet and negatively correlated with fibre intake [47].

The state of intestinal flora is a key factor for elucidating the mechanisms of TCM efficacy. Correlation analysis of the microbiota with metabolites showed that *Bacillus* and *Solibacillus* were strongly associated with different metabolites (7 days). *Bacillus* is a probiotic widely used in dairy farming [48]. Jia et al. [49,50] indicated that feeding *Bacillus subtilis* increases dry matter intake and milk production, reduces intestinal CH_4_ emissions, and affects rumen microbial activity in dairy cows. The increased relative abundance of *Solibacillus* is considered an adaptive response to increased dietary diversity [51]. The active ingredients of DBT, including ferulic acid and feruloylated oligosaccharides, can increase the relative abundance of *Solibacillus* and other genera of bacteria and modulate the gut microbiota and microbial metabolism to alleviate anxiety and depression [52]. In contrast, at 14 days postpartum, the relative abundance of Bacteroidetes was significantly increased in the H-DBT group. A change in the relative abundance of Firmicutes was not observed, leading to a decrease in the F/B ratio. Bacteroidetes bacteria contain several enzymes essential for carbohydrate metabolism in dairy cows and can synthesise conjugated linoleic acid, which is involved in lipid metabolism and immunomodulation [43,53]. *Bacteroides intestinalis* can convert primary bile acids into secondary bile acids, which are reabsorbed by chylomicrons and enter the lipid metabolism pathway [54]. It has been shown that feeding citrus flavonoid extracts (CFE) results in notably higher levels of the phylum Bacteroidota and genus *Bacteroides* in the faeces of dairy cows than those in the control. It was inferred that the increase in the *Bacteroides* proportion is associated with the health benefits of CFE [55]. Correlation analysis of the microbiota with metabolites showed that *Dorea* and *Romboutsia* were associated with different metabolites (14 days). *Dorea* partially predicts weight loss, and a high *Dorea* abundance may be associated with difficulty losing excess weight due to calorie restriction [56]. It has been shown that *Lactobacillus rhamnosus* hsryfm 1301 reduces the relative abundance of intestinal *Dorea* in hyperlipidaemic mice and that *Dorea* is significantly positively correlated with serum lipids [57]. *Dorea* can interact with bile acids to complete biotransformation processes, such as 7-dehydroxylation and 3-epimerization [58]. Several research groups have shown that *Romboutsia* is an obesity-related phylotype, which is positively correlated with the indicators of lipid metabolism [59,60,61]. *Romboutsia*, a member of the Peptostreptococcaceae family, is associated with lipid metabolic pathways, including cholesterol and glycerolipid metabolism [62]. In hyperketonaemic cows, *Romboutsia* is correlated with a wide range of bile acids; it has been suggested that this is related to the fact that *Romboutsia* has a specific gene conjugated with a bile acid hydrolase [63]. Our study revealed that *Dorea* and *Romboutsia* were significantly correlated with several BAs, particularly glycine-conjugated BA, suggesting that they may influence the binding of bile acids to glycine and, hence, bile acid metabolism.

Changes in the plasma metabolic profiles of dairy cows at 7 and 14 days postpartum under DBT treatment were analysed using targeted metabolomics. Notably, acetylcarnitine is the only metabolite that changed at both time points, with a significant decrease at 7 days postpartum (*p* < 0.05) and a decreasing trend at 14 days postpartum (*p* < 0.1). Acetylcarnitine is essential in the process of fatty acid oxidation. In the cytoplasm, NEFAs undergo activation to form acyl-CoA. This acyl-CoA is then transported into the mitochondria through the carnitine shuttle system, where it is converted into one molecule of acetyl-CoA that undergoes a two-carbon chain shortening [64,65]. Acetyl-CoA is involved in four major metabolic pathways [66]: (1) acetyl-CoA and free carnitine are converted into acetylcarnitine by carnitine acetyltransferase; (2) acetyl-CoA participates in the TCA cycle; (3) acetyl-CoA is a substrate for cholesterol synthesis; (4) acetyl-CoA is converted into a ketone body. Acetylcarnitine concentrations are significantly increased in cows with high lipolysis; it was concluded that high levels of acetyl-CoA produced by the oxidation of NEFA in cows with high lipolysis exceed the capacity of the TCA and increase their conversion to acetylcarnitine [5,67]. Furthermore, acetylcarnitine positively correlates with the serum NEFA concentration [68]. Dyslipidaemia in type 2 diabetes mellitus (T2DM) is associated with high acetylcarnitine levels [69,70]. In our study, the reduction in acetylcarnitine at two time points suggests that DBT may alleviate lipid mobilisation in postpartum dairy cows. 2-HB is a metabolic biomarker for various diseases, including insulin sensitivity, T2DM, major cardiovascular diseases, and cancer, and is involved in lipid oxidation and oxidative stress [71,72]. 2-HB is synthesised as a co-product of protein metabolism during insulin resistance (IR), which is intrinsically related to an increase in 2-HB levels, and decreases with the improvement of insulin resistance [73,74]. 2-HB is also closely related to lipid metabolism; ref. [75] showed that 2-HB regulates lipid metabolism both in vivo and in vitro. Consistent with the serum phenotypic index results, a reduction in the BHB concentration was also observed in plasma-targeted metabolomics. Incomplete oxidation of NEFAs is the main source of ketones. BHB is the predominant form for metabolic use among the three ketone bodies, with higher transport effectiveness and better circulating concentrations [76]. However, BHB accumulation in the blood can cause hyperketonaemia, which harms cow health and increases the risk of other diseases [77]. Therefore, we suggest that DBT regulation of lipid metabolism in dairy cows is closely related to a decrease in acetylcarnitine, 2-HB, and BHB contents in the blood. DBT infusion significantly increases the concentrations of malic and fumaric acids (7 days), which are intermediates of the TCA cycle. It has been shown that DBT may exert anti-fatigue effects by modulating the TCA cycle [19]. The status of the TCA cycle serves as a comprehensive reflection of energy metabolism within the body [78]. KEGG enrichment analyses and differential metabolite pathway maps indicate that DBT can enhance TCA cycling, improving the energy status of dairy cows in the postpartum period.

In addition to lipid digestion, BA may act as a signalling molecule to influence Glu and lipid metabolism [6,79]. Gu et al. [80] reported the significant activation of secondary BA biosynthesis and significant changes in BA profiles in postpartum cows with excessive lipolysis. Here, we found that DBT significantly changed the plasma bile acid profile of dairy cows at 14 days postpartum, especially the glycine-conjugated BA, including GUDCA, GDCA, and GCDCA. *Dorea* and *Romboutsia* may influence the binding of bile acids to glycine and, hence, bile acid metabolism. Furthermore, BHB is significantly and positively correlated with several bile acids. This is consistent with previous findings on hyperketonaemia and non-hyperketonaemia in dairy cows in the early postpartum period, wherein BA concentrations were similar in both groups and BA and microbiota were correlated with plasma BHB concentrations [63]. Although it has been shown that increased lipid mobilisation in over-conditioned postpartum cows was accompanied by lower circulating BA concentrations, it may be due to higher faecal BA excretion [8]. Our previous study showed that BA derived from gut microbes is altered during the development of hyperketonaemia; GUDCA is an important functional readout involved in lipid mobilisation in periparturient dairy cows [44]. Moreover, GCDCA is the main toxic component of BA in the bile and serum of patients with cholestasis [81], directly inducing apoptosis and necrosis of hepatocytes [82]. Although recent studies have shown that GUDCA and GDCA treatments can alter BA levels and intestinal flora to ameliorate disease [83,84], further studies are needed to investigate their role and causality in lipid mobilisation in periparturient dairy cows. Overall, our experimental results suggest that DBT can regulate lipid metabolism in post-parturient dairy cows by altering the intestinal microbiota and influencing serum lipid metabolites, including BHB, acetylcarnitine, 2-HB, and bile acid profiles.

## 5. Conclusions

In this study, 16S rRNA sequencing and metabolomics were integrated to investigate the role of DBT in postpartum cows, especially in lipid metabolism. DBT was found to improve lipid metabolism by modulating blood concentrations of BHB, acetylcarnitine, and 2-HB. DBT also regulated the plasma bile acid profile, particularly by decreasing glycine-conjugated bile acids. 16S rRNA sequencing was used to analyse the dynamics of the intestinal flora, showing that DBT modulated the composition and structure of the hindgut microbiota. Correlations were observed between differential microbiota profiles and lipid metabolites, which are crucial for DBT’s regulation of lipid metabolism. However, limitations exist in this study. A future study should include targeted lipid metabolomics on plasma samples from cows post-DBT feeding to better understand changes in lipid metabolic profiles and the underlying mechanisms. This study provides new insights into the role of DBT in regulating lipid mobilisation in postpartum dairy cows. It supports the clinical application of DBT in dairy farming, offering a potential alternative to antibiotics and promoting more sustainable farming practices.

## Figures and Tables

**Figure 1 metabolites-15-00058-f001:**
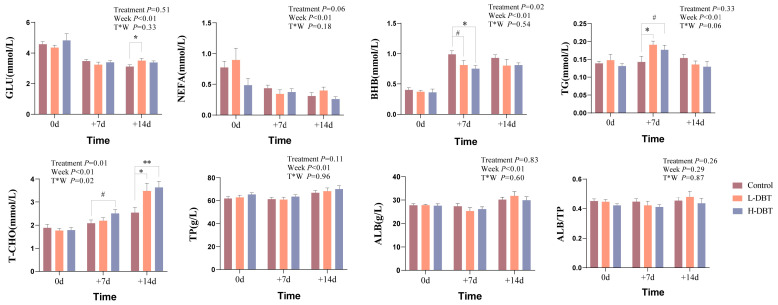
Blood biochemical indices of the control, L-DBT, and H-DBT groups. Horizontal and vertical coordinates show the different time points and measured indices, respectively. Brown, yellow, and blue columns represent the control, L-DBT, and H-DBT groups, respectively (0.05 < ^#^
*p* < 0.1; * *p* < 0.05; ** *p* < 0.01).

**Figure 2 metabolites-15-00058-f002:**
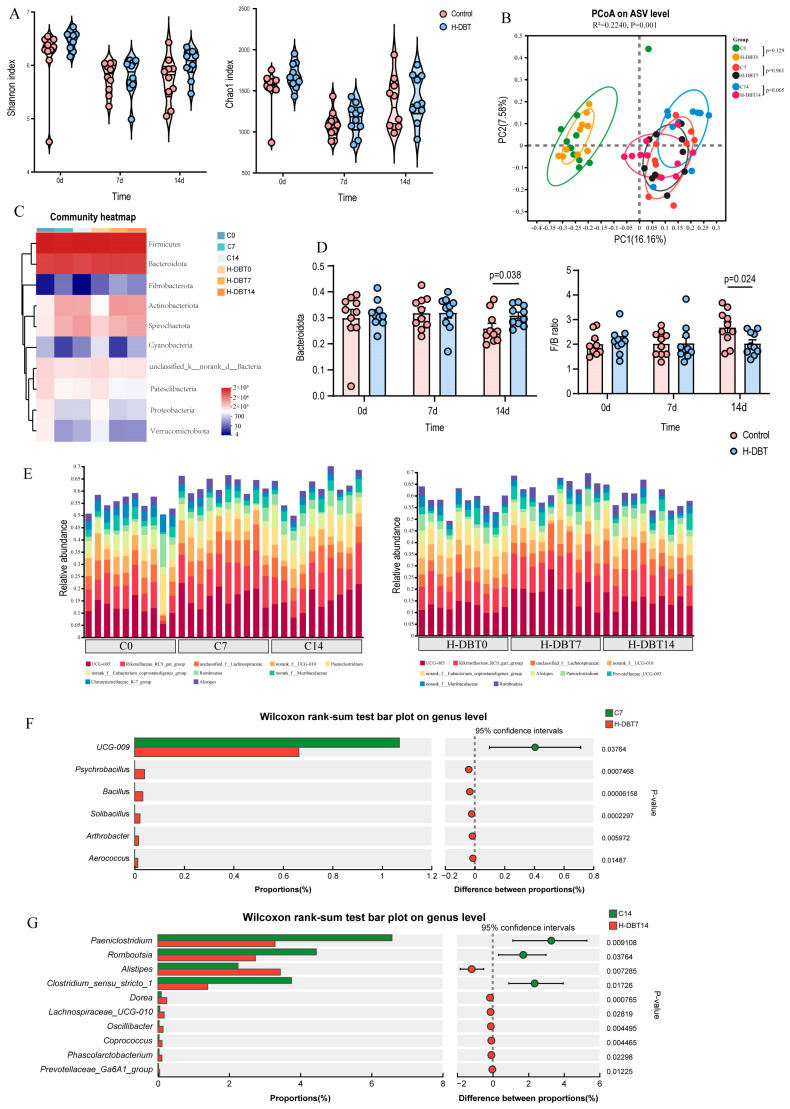
DBT-altered faecal microbial community of dairy cows during the late periparturient period. (**A**) Alpha diversity of faecal microbiota, including Shannon and Chao1 indices, at different time points. (**B**) PCoA plot of the beta diversity based on the Bray–Curtis distance. (**C**) Community heatmap of faecal microbiota at the phylum level. (**D**) Relative abundance of Bacteroidetes and the ratio of Firmicutes to Bacteroidetes (F/B). Stacked bar chart (**E**) of the top 10 species with relative abundance of faecal microbiota at the genus level. Each stacked bar chart represents one sample. (**F**) Differential faecal bacteria at the genus level (7 days). (**G**) Top 10 differential faecal bacteria at the genus level (14 days).

**Figure 3 metabolites-15-00058-f003:**
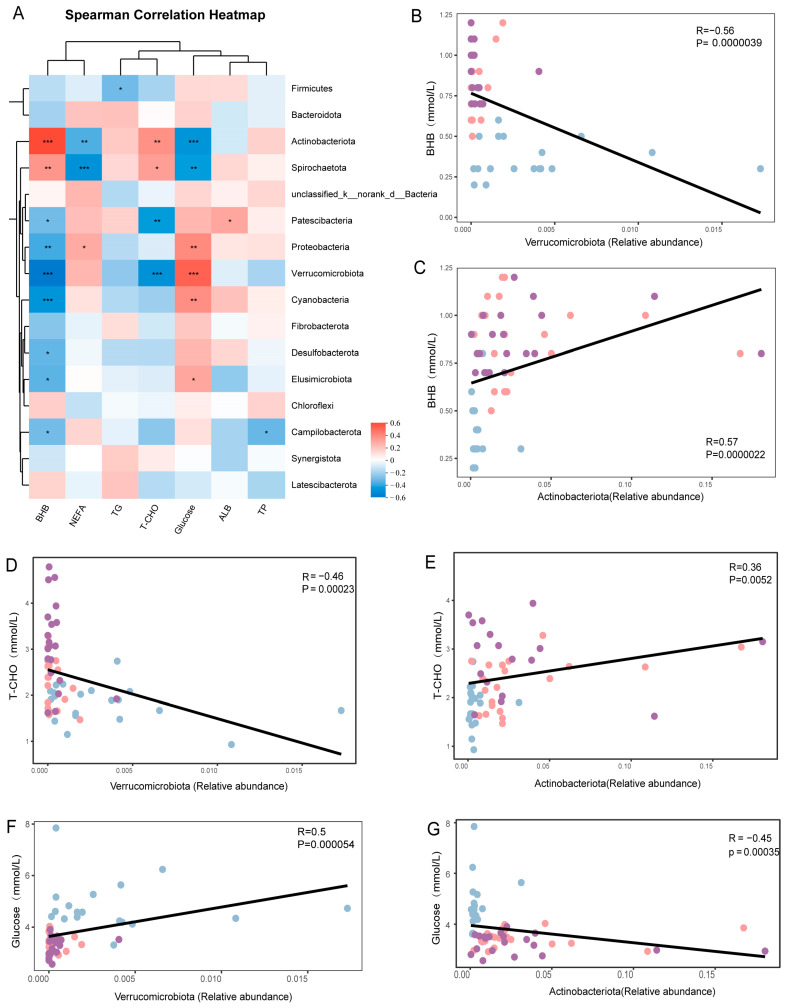
Correlation of faecal microbiota at the phylum level with clinical phenotype indices at three time points. * *p* < 0.05, ** *p* < 0.01, and *** *p* < 0.001. (**A**) Spearman’s correlation analysis between microbiome and phenotypic indicators. (**B**–**G**) Associations between the abundance of Verrucomicrobiota or Actinobacteriota and circulating BHB, T-CHO, or glucose concentrations in dairy cows based on Spearman’s correlation analysis. The blue dots represent 0 day samples, the orange dots represent +7 days samples, and the purple dots represent +14 days samples.

**Figure 4 metabolites-15-00058-f004:**
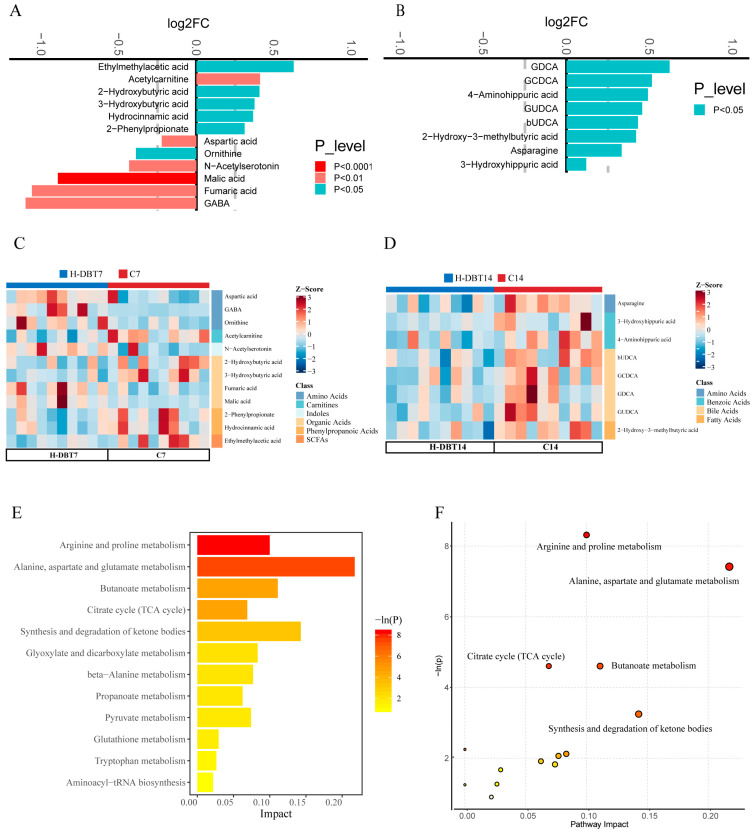
Significantly differential metabolites and KEGG enrichment metabolic pathways. (**A**) Differential metabolite bar graph (7 days). Each metabolite in the figure corresponds to a horizontal bar. The colour and the length of the horizontal bar represent the *p* and log2FC values, respectively. (**B**) Differential metabolite bar graph (14 days). (**C**) Differential metabolite heatmap (7 days). (**D**) Differential metabolite heatmap (14 days). (**E**) Enrichment analysis bar graph (7 days). Each pathway in the figure corresponds to a horizontal bar. The colour and the length of the horizontal bar represent the −ln(*p*) and impact values, respectively. (**F**) Bubble diagram for pathway enrichment analysis (7 days). The horizontal coordinate indicates the pathway impact. The size of the circle is related to the pathway impact. The larger the impact value is, the larger is the circle. The vertical coordinate represents the −ln(*p*) obtained from the pathway enrichment analysis. The yellow–red colour change of the point is positively correlated with the negative logarithm of the *p*-value of the change in pathway. Pathways with *p* < 0.05 are labelled with a name in the figure, and pathways that do not meet the abovementioned conditions are not labelled in the figure.

**Figure 5 metabolites-15-00058-f005:**
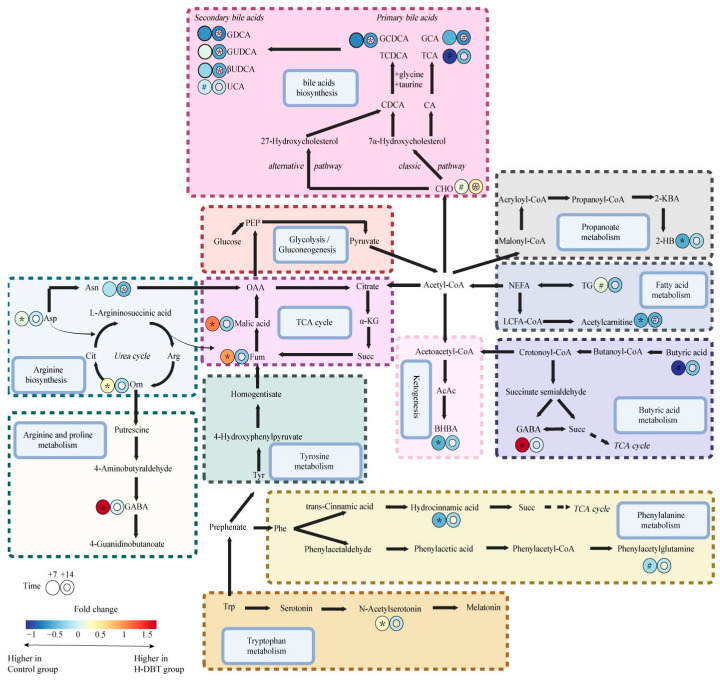
Differential metabolite pathway diagram. The circle colour indicates the fold change between control and H-DBT groups at each time point. Cit, citrulline; Arg, arginine; Orn, ornithine; Fum, fumaric acid; OAA, oxaloacetate; α-KG, α-ketoglutaric acid; Succ, succinate; AcAc, acetoacetic acid; 2-KBA, 2-ketobutyric acid; Tyr, tyrosine; Phe, phenylalanine; LCFA-CoA, long-chain-fatty acyl-CoA; Trp, tryptophan. ^#^ 0.05 < *p*  <  0.1; * *p*  <  0.05.

**Figure 6 metabolites-15-00058-f006:**
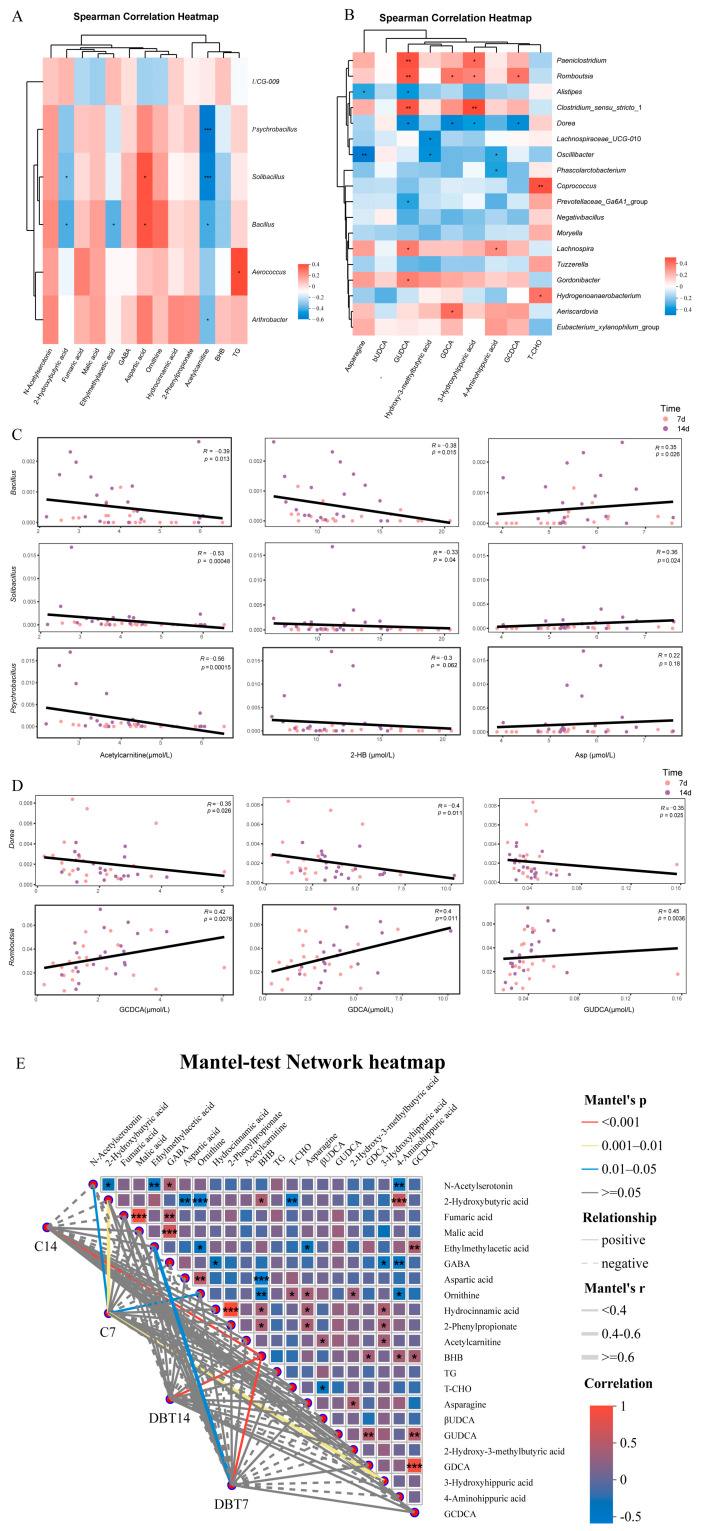
Correlation among phenotypes, genera, and serum metabolites. (**A**) Spearman’s correlation analysis of differential metabolites and relative abundance of differential genera (7 days). * *p* < 0.05, *** *p* < 0.001. (**B**) Spearman’s correlation analysis of differential metabolites and relative abundance of differential genera at the genus level (14 days). * *p* < 0.05, ** *p* < 0.01. (**C**) Scatterplot of associations between the relative abundance of *Psychrobacillus*, *Bacillu*, or *Solibacillus* and circulating acetylcarnitine, 2-HB, and Asp levels at two time points based on Spearman’s correlation analysis. (**D**) Scatterplot of associations between the relative abundance of *Dore* or *Romboutsia* and circulating GCDCA, GDCA, and GUDCA levels at two time points based on Spearman’s correlation analysis. (**E**) Mantel-test network heatmap. Links between gut microbial community and host phenotypes. The heatmap displays the relationships among differential metabolites based on Spearman’s correlation analysis. The line indicates the relationship of the gut microbiota (genus level) matrix with the blood variables matrix based on the Mantel test. The line colour indicates the *p*-value of the Mantel test, and the thickness of the line indicates the correlation coefficient. * *p* < 0.05, ** *p* < 0.01, and *** *p* < 0.001.

**Figure 7 metabolites-15-00058-f007:**
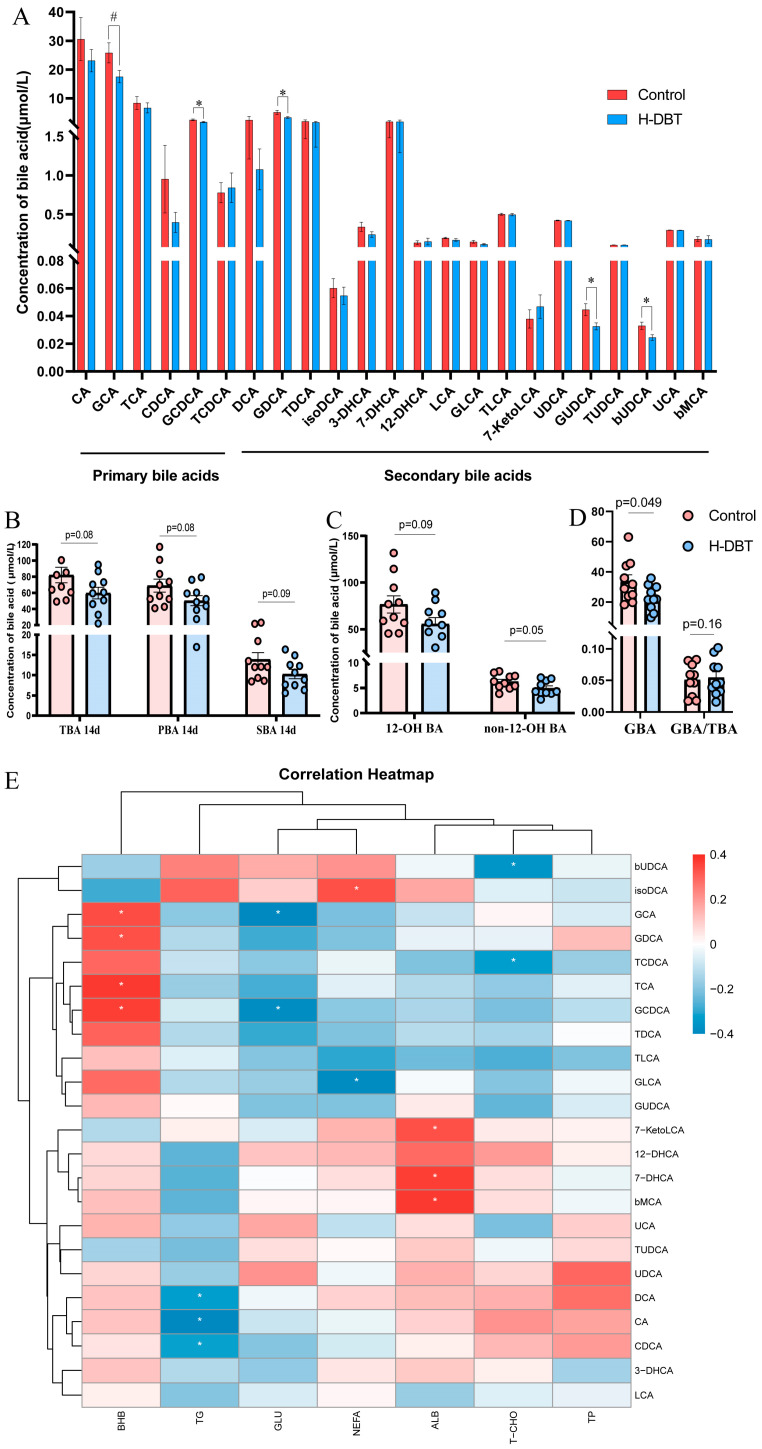
Bile acid metabolic profiles of plasma at 14 days after calving. (**A**) The bar graph shows the variation in 23 bile acids in plasma. # *p*< 0.1, * *p* < 0.05. CA, cholic acid; GCA, glycocholic acid; TCA, taurocholic acid; CDCA, taurocholic acid; GCDCA, glycochenodeoxycholate; TCDCA, taurochenodeoxycholate; DCA, deoxycholic acid; GDCA, glycodeoxycholic acid; TDCA, taurodeoxycholate; iso-DCA, isodeoxycholic acid; 3-DHCA, 3-dehydrocholic acid; 7-DHCA, 7-dehydrocholic acid; 12-DHCA, 12-dehydrocholic acid; LCA, lithocholic acid; GLCA, glycolithocholate; TLCA, taurolithocholic acid sulfate; 7-ketoLCA, 7-ketolithocholic acid; UDCA, ursodeoxycholic acid; GUDCA, glycoursodeoxycholic acid; TUDCA, tauroursodeoxycholic acid; bUDCA, 3β-ursodeoxycholic acid; UCA, ursocholic acid; bMCA, β-muricholic acid. (**B**) The concentration of total bile acid (TBA), primary bile acid (PBA), and secondary bile acid (SBA). (**C**) The concentration of 12-OH and non-12-OH BA. (**D**) The concentration of glycine-conjugated BA (GBA) and the ratio of GBA to TBA. (**E**) Spearman’s correlation analysis between 23 bile acids and phenotypic indicators. * *p* < 0.05.

**Table 1 metabolites-15-00058-t001:** Results of the chi-square test (+7 days).

Comparison	Number of Cows with BHB ≥ 1 mmol/L	Total	χ^2^	*p*
Control	7	10	5.051	0.025 > 0.0167
L-DBT	2	10		
Control	7	10	7.500	0.006 < 0.0167
H-DBT	1	10		
L-DBT	2	10	0.392	0.531 > 0.0167
H-DBT	1	10		

**Table 2 metabolites-15-00058-t002:** Results of the chi-square test (+14 days).

Comparison	Number of Cows with BHB ≥ 1 mmol/L	Total	χ^2^	*p*
Control	5	10	0.833	0.361 > 0.0167
L-DBT	3	10		
Control	5	10	3.810	0.051 > 0.0167
H-DBT	1	10		
L-DBT	3	10	1.250	1.250 > 0.0167
H-DBT	1	10		

## Data Availability

The datasets generated in this study have been deposited in online repositories. The repository name and accession number are as follow: NCBI-PRJNA1206075.

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
