# Peer review of "The Positive Regulatory Effect of DBT on Lipid Metabolism in Postpartum Dairy Cows"

_metabolites, 2025, doi:10.3390/metabo15010058_

Round 1

Reviewer 1 Report

Comments and Suggestions for Authors

Line 131-133: Those lines are difficult to understand, what does “200 g HQ, 40 g DG, 3000 mL water/each time” mean?, please specify the preparation of DBT solution and the feeding method, mixed it with TMR or ...?. In addition, there is difference in average feed intake among groups?, also, TMR composition should be provided.

Line 135: How the authors collected fecal samples?

Line 151: PCR cycling conditions are needed, where is the raw sequence data deposited and what is the accession number?

Line 165: UPLC-MS/MS conditions should be provided.

Line180: Partition of chi-square was performed based on the number of BHB ≥ 1. what does the “the number of BHB ≥ 1” mean?, what is the reason for authors to select it.

Line246-247, line282-283, line376-377: Should “DBT” in Figure 2, Figure 4 and Figure 7 be “H-DBT” ?.

Line 524-537: Please summarize specific findings instead of methodology descriptions, please provide some recommendations for farmers how to use DBT in more details.

Author Response

Response to Reviewer 1 Comments

Thank you for taking the time to review this manuscript. Please find the detailed responses below, with the corresponding revisions and corrections highlighted or using track changes in the re-submitted files.

Point-by-point response to Comments and Suggestions for Authors

Comments 1: Those lines are difficult to understand, what does “200 g HQ, 40 g DG, 3000 mL water/each time” mean?, please specify the preparation of DBT solution and the feeding method, mixed it with TMR or ...?. In addition, there is difference in average feed intake among groups?, also, TMR composition should be provided.

Thanks for your comment. The preparation of DBT has been noted in the manuscript (Page3, lines 120-127):

HQ and DG were purchased from Minxian Ronghe Pharmaceutical Co., Ltd. (Gansu, Dingxi, China). Each herb was air-dried and ground in a mixer to a fine powder (particle size <15 µm). Fine powders of HQ and DG were dissolved in 3000 mL of hot water at a ratio of 5:1 and allowed to cool.

We have modified “200 g HQ, 40 g DG, 3000 mL water/each time” to: Thirty cows were randomly divided into three groups (10 cows in each group): high-dose DBT group (H-DBT group), low-dose DBT group (L-DBT group) and control group. Cows in the H-DBT group received a daily dose of 240g DBT (200 g HQ, 40 g DG), cows in the L-DBT group received a daily dose of 120g DBT (100 g HQ, 20 g DG), and cows in the control group received 3,000 mL of water per day like the others. Treatments were administered via esophageal tubing inserted directly into the rumen through a stainless steel speculum. The placement of the distal end of the tubing in the rumen liquor was confirmed by blowing on the proximal end of the tubing and listening for gas bubbles using a stethoscope, ensuring accurate delivery of the treatment directly into the rumen. This change in the revised manuscript can be found in Page 3-4, lines 139-148.

We have modified it to: Dry matter intakes (DMI) were calculated based on the amounts of feed offered and refused, together with the DM content of the TMR. There was no difference in average DMI among groups. This change in the revised manuscript can be found in Page 3, lines 134-137.

The DMI date added to supplementary material

Supplementary Table S2 Dry matter intake in postpartum dairy cows

Item

Testing time

Group

Control

L-DBT

H-DBT

DMI

(kg)

0d

7.17

7.04

7.21

+7d

17.59

17.83

17.81

+14d

18.03

18.09

18.13

The TMR data added to supplementary material

Supplementary Table S1 Ingredients and chemical composition of the diets for dairy cows

Item

DM basis, %

Item

DM basis, %

complete feeds

42.74

DM

49.91

fat powder

1

CP

18.48

molasses

3.76

NDF

30.5

corn silage

26.59

ADF

17.53

alfalfa hay

15.23

Ca

0.83

oat hay

2.32

P

0.45

beet pulp

3.57

NEL, Mcal/kg of DM

1.74

baking soda

0.25

cottonseed

4.54

Note: Guaranteed value (%) of composition of each kilogram of premix: crude protein≥24.8, crude fat≥4.66, crude fiber≤5.34, calcium≥0.92, phosphorus ≥0.4, lysine≥ 0.35, sodium chloride≥1. The net energy of lactation is the calculated value, and others are the measured value. DM=dry matter; CP= crude protein; NDF= neutral detergent fiber; ADF= acid detergent fiber; Ca= calcium; P=phosphorus; NEL=net energy of lactation.

Comments 2: How the authors collected fecal samples?

Thank you for pointing this out. Faecal samples were collected via the rectum using disposable sterile long arm gloves, transferred into freezing tubes and stored at -80°C.

This change in the revised manuscript can be found in Page 4, lines 158-160.

Comments 3: PCR cycling conditions are needed, where is the raw sequence data deposited and what is the accession number?

PCR amplification cycling conditions were as follows: initial denaturation at 95 ℃ for 3 min, followed by 27 cycles of denaturation at 95 ℃ for 30 s, annealing at 55 ℃ for 30 s and extension at 72 ℃for 45 s, with a final extension at 72 ℃ for 10 min, followed by cooling to 4 ℃. This revision can be found on Page 4, lines 179-182.

The datasets generated in this study have been deposited in online repositories. The repository name and accession number are as follow: NCBI-PRJNA1206075. (link: https://www.ncbi.nlm.nih.gov/sra/PRJNA1206075)

This change in the revised manuscript can be found on Page 19, lines 581-583.

Comments 4:  UPLC-MS/MS conditions should be provided.

Thanks for your comment. The statement has been noted in the manuscript (Page5 lines 194-197):Specific sample preparation methods, details on the mobile phase and gradient elution procedure for UPLC, and mass spectrometry conditions are described in the supplementary materials.

Here are the details.

Plasma sample (20 μL) were thoroughly mixed containing 120 μL of ice-cold methanol containing partial internal standards and then centrifuged at 4 °C (4000 g, 30 min). Supernatant (30 μL) was collected, derivative reagents (20 μL) were adde. The mixture was sealed, and derivatization was carried out at 30 °C for 60 min. After derivatisation, the sample was diluted with 330 μL of ice-cold 50% methanol solution, stored at -20 °C for 20 min and centrifuged at 4 °C (4000 g, 30 min). The supernatant (135 μL) was pipetted and 10 μL of internal standard was added. Serial dilutions of derivatised stock standards were added to the remaining wells. An ultra-performance liquid chromatography–tandem mass spectrometry (UPLC-MS/MS) system (ACQUITY UPLC-Xevo TQ-S, Waters Corp., Milford, MA, USA) was used to quantify all targeted metabolites. The system was equipped with an ACQUITY UPLC BEH C18 1.7 µM VanGuard pre-column (2.1 × 5 mm) and ACQUITY UPLC BEH C18 1.7 µM analytical column (2.1 × 100 mm), with an injection volume of 5 µL and flow rate of 0.4 mL/min. The column temperature was maintained at 40 °C. The mobile phase consisted of A (water with 0.1% formic acid) and B (acetonitrile/IPA; 70:30). The gradient elution procedure was as follows: 0–1 min (5% B), 1–5 min (5–30% B), 5–9 min (30–50% B), 9–11 min (50–78% B), 11–13.5 min (78–95% B), 13.5–14 min (95–100% B), 14–16 min (100% B), 16–16.1 min (100–5% B), and 16.1–18 min (5% B). The MS was alternately operated in positive-ion mode (+1.5 kV) and negative-ion mode (−2 kV). The source temperature was set to 150 °C, the desolvation temperature was set to 550 °C, and the desolvation gas flow was set to 1000 L/h. Raw data generated by UPLC-MS/MS were processed using MassLynx software (v4.1; Waters, Milford, MA, USA) for peak integration, calibration, and quantification of each metabolite. The iMAP platform (v1.0; MetaboProfile, Shanghai, China) was used for the statistical analyses. The concentrations of targeted metabolites were determined based on the calibration curves and corresponding regression coefficients.

Comments 5: Partition of chi-square was performed based on the number of BHB ≥ 1. what does the “the number of BHB ≥ 1” mean?, what is the reason for authors to select it.

Sorry for the misunderstanding, we have revised the description to be clearer.

Chi-square was performed based on the number cows with serum concentrations BHB ≥ 1 mmol/L. This change in the revised manuscript can be found in Page 5, lines 206-207.

The threshold values for interpreting BHB were based on previous research; BHB ≥1 mmol/L adversely affects reproductive performance and milk production and is associated with an increased risk of subsequent diseases.

References 34 and 35 in the revised manuscript:

[34] Ospina, P.A.; Nydam, D.V.; Stokol, T.; Overton, T.R. Associations of elevated nonesterified fatty acids and β-hydroxybutyrate concentrations with early lactation reproductive performance and milk production in transition dairy cattle in the northeastern United States. Journal of Dairy Science 2010, 93, 1596-1603, doi:10.3168/jds.2009-2852.

[35] Ospina, P.A.; Nydam, D.V.; Stokol, T.; Overton, T.R. Evaluation of nonesterified fatty acids and β-hydroxybutyrate in transition dairy cattle in the northeastern United States: Critical thresholds for prediction of clinical diseases. Journal of Dairy Science 2010, 93, 546-554, doi:10.3168/jds.2009-2277.

  and BHB ≥ 1mmol/L has been used as a marker of NEB in postpartum period dairy cows. This change in the revised manuscript can be found in Page 16, lines 425-426.

References 36 and 37 in the revised manuscript:

[36] Macrae, A.I.; Burrough, E.; Forrest, J.; Corbishley, A.; Russell, G.; Shaw, D.J. Prevalence of excessive negative energy balance in commercial United Kingdom dairy herds. Veterinary Journal 2019, 248, 51-57, doi:10.1016/j.tvjl.2019.04.001.

[37] Ospina, P.A.; McArt, J.A.; Overton, T.R.; Stokol, T.; Nydam, D.V. Using Nonesterified Fatty Acids and β-Hydroxybutyrate Concentrations During the Transition Period for Herd-Level Monitoring of Increased Risk of Disease and Decreased Reproductive and Milking Performance. Veterinary Clinics of North America-Food Animal Practice 2013, 29, 387-+, doi:10.1016/j.cvfa.2013.04.003

Comments 6: Line246-247, line282-283, line376-377: Should “DBT” in Figure 2, Figure 4 and Figure 7 be “H-DBT” ?.

Thank you for pointing this out. We have changed the DBT in Figure 2, Figure 4 and Figure 7 to H-DBT.

Comments 7: Line 524-537: Please summarize specific findings instead of methodology descriptions, please provide some recommendations for farmers how to use DBT in more details.

Thank you for your suggestions. We have revised the manuscript accordingly: In this study, 16S rRNA sequencing and metabolomics were integrated to investigate the role of DBT in postpartum cows, especially in lipid metabolism. DBT was found to improve lipid metabolism by modulating blood concentrations of BHB, acetylcarnitine, and 2-HB. DBT also regulated plasma bile acid profile, particularly by decreasing glycine-conjugated bile acids. 16S rRNA sequencing was used to analyse the dynamics of the intestinal flora, showing that DBT modulated the composition and structure of the hindgut microbiota. Correlations were observed between differential microbiota profiles and lipid metabolites, which are crucial for DBT’s regulation of lipid metabolism. However, limitations exist in this study. Future study should include targeted lipid metabolomics on plasma samples from cows post-DBT feeding to better understand changes in lipid metabolic profiles and the underlying mechanisms. This study provides new insights into the role of DBT in regulating lipid mobilization in postpartum dairy cows. It supports the clinical application of DBT in dairy farming, offering a potential alternative to antibiotics and promoting more sustainable farming practices.

This change in the revised manuscript can be found in Page 18-19, lines 553-566.

Reviewer 2 Report

Comments and Suggestions for Authors

Abstract

L29 Break the sentence

Material and methods

How was the DBT administered to the cow, please include this information

L145 Provide details of the experimental process for biochemical indices rather than stating you followed manufacturer protocol. What was involved?

L195 Correct and Replace with found

L194 HDBT7???? What does HDBT7 OR 14 mean, you defined HDBT but not HDBT 7 OR 14. If you mean 7 and 14 days, please be explicit

Results

3.1 No mention of GLU

L265 Please rephrase for clarity

Author Response

Response to Reviewer 2 Comments

Thank you very much for taking the time to review this manuscript. Please find the responses below.

Point-by-point response to Comments and Suggestions for Authors

Comments 1: L29 Break the sentence

Thank you, we have revised it: Our research shows that, in dairy cows 7 days postpartum, DBT significantly reduced serum 3-Hydroxybutyric acid (BHB) concentrations and the number of cows with BHB concentrations ≥1 mmol/L. Additionally, DBT increased serum total cholesterol contents at both 7 and 14 days postpartum. This change in the revised manuscript can be found in Page 1, lines38-41.

Comments 2: How was the DBT administered to the cow, please include this information

Thank you, we have modified the manuscript as follows: Treatments were administered via esophageal tubing inserted directly into the rumen through a stainless steel speculum. The placement of the distal end of the tubing in the rumen liquor was confirmed by blowing on the proximal end of the tubing and listening for gas bubbles using a stethoscope, ensuring accurate delivery of the treatment directly into the rumen. This change can be found in Page 4, lines 144-148.

Reference:M.M. Pickett.; M.S. Piepenbrink.; T.R. Overton. Effects of Propylene Glycol or Fat Drench on Plasma Metabolites, Liver Composition, and Production of Dairy Cows During the Periparturient Period. Journal of Dairy Science 2003, 86, 2113-2121, doi:10.3168/jds.s0022-0302(03)73801-6

Comments 3: L145 Provide details of the experimental process for biochemical indices rather than stating you followed manufacturer protocol. What was involved?

Thank you for pointing this out. We have modified it to: GLU was measured using the glucose oxidase method; NEFA was determined by enzymatic assay; TG was measured using the GPO-PAP enzymatic method; T-CHO was quantified via the COD-PAP enzymatic method; TP was assessed using the BCA microplate method; ALB was measured using microwell plate method and GSK3B was measured using a competitive method (antigen-antibody binding). This change in the revised manuscript can be found in Page 4 lines 169-173.

The instructions can be found in official website of Nanjing Jiancheng Bioengineering Institute (www.njjcbio.com)

Comments 4: L195 Correct and Replace with found

Thanks for your comment. We have modified it to: The effects of DBT on the blood biochemical indicators of postpartum dairy cows are illustrated in Figure 1. The results indicate that, the H-DBT7 group exhibited significantly reduced BHB values compared with those of the control group (p < 0.05; Fig. 1); moreover, a trend towards lower BHB values was observed for the L-DBT7 group (0.05 < p <0 .1). This change can be found on Page 5, lines 221-222.

Comments 5:L194 HDBT7???? What does HDBT7 OR 14 mean, you defined HDBT but not HDBT 7 OR 14. If you mean 7 and 14 days, please be explicit

Thanks for your comment. We have modified it to: The control and H-DBT groups, on 0 days, were named C0 and H-DBT0, respectively. The control and H-DBT groups, on +7 days, were named C7 and H-DBT7, respectively. The control and H-DBT groups, on +14 days, were named C14 and H-DBT14, respectively. This change in the revised manuscript can be found in Page 4 lines 152-155.

Comments 6:3.1 No mention of GLU

Thank you for pointing this out. We have modified it to: Serum GLU levels in the L-DBT14 group significantly elevated compared to the control group (p < 0.05). This change in the revised manuscript can be found in Page 5 lines 226-227.

Comments 7:L265 Please rephrase for clarity

Thank you for pointing this out. We have modified it to Subsequently, we queried differential metabolites in KEGG pathway database and searched relevant literature for data relating to overall metabolism pathway (Fig. 5). This change in the revised manuscript can be found in Page 11 lines 333-334.

Round 2

Reviewer 1 Report

Comments and Suggestions for Authors

No additional comments